# Neurotoxicity of Chronic Co-Exposure of Lead and Ionic Liquid in Common Carp: Synergistic or Antagonistic?

**DOI:** 10.3390/ijms23116282

**Published:** 2022-06-03

**Authors:** Weikai Ding, Yousef Sultan, Shumei Li, Wenjun Wen, Bangjun Zhang, Yiyi Feng, Junguo Ma, Xiaoyu Li

**Affiliations:** 1Henan International Joint Laboratory of Aquatic Toxicology and Health Protection, College of Life Science, Henan Normal University, Xinxiang 453007, China; 1804083042@stu.htu.edu.cn (W.D.); yy.abdel-rahim@nrc.sci.eg (Y.S.); 1904283092@stu.htu.edu.cn (S.L.); 2104183032@stu.htu.edu.cn (W.W.); 041129@htu.edu.cn (B.Z.); 2020040@htu.edu.cn (Y.F.); 2016023@htu.edu.cn (J.M.); 2Food Toxicology and Contaminants Department, National Research Centre, Dokki, Cairo 12622, Egypt

**Keywords:** *Cyprinus carpio*, Pb, ionic liquid, combined exposure, neurotoxicity

## Abstract

Previous studies have indicated that the harmful heavy metal lead (Pb) contamination in aquatic systems has caused intelligence development disorders and nervous system function abnormalities in juveniles due to the increased permeability of the blood–brain barrier. Ionic liquids (ILs) are considered “green” organic solvents that can replace traditional organic solvents. Studies have found the presence of ILs in soil and water due to chemical applications or unintentional leakage. Therefore, what would happen if Pb interacted with ILs in a body of water? Could ILs enable Pb to more easily cross the blood–brain barrier? Therefore, we examined the combined exposure of Pb and ILs in common carp at low concentration (18.3 mg L^−1^ of Pb(CH_3_COO)_2_•3 H_2_O and 11 mg L^−1^ of the IL 1-methyl-3-octylimidazolium chloride, 5% of their LC_50_) for 28 days in the present study. The result of a neurobehavioral assay showed that chronic exposure of lead at lower concentrations significantly altered fish movement and neurobehaviors, indicating that lead exposure caused neurotoxicity in the carp. Increases in the neurotransmitter dopamine levels and injuries in the fish brain accounted for neurobehavioral abnormalities induced by lead exposure. Moreover, we also found that lead could easily cross the blood–brain barrier and caused significant bioaccumulation in the brain. Particularly, our study indicated that the ionic liquid could not synergistically promote blood–brain barrier permeability and hence failed to increase the absorption of lead in the fish brain, suggesting that the combined exposure of lead and ILs was not a synergistic effect but antagonism to the neurotoxicity. The results of this study suggested that ILs could recede the Pb induced neurotoxicity in fish.

## 1. Introduction

Heavy metal lead (Pb) is a toxic element to living beings, and it is present in various environments due to industrial Pb release, expired batteries, the use of Pb paint and fuel, etc. [1,2,3]. The World Health Organization (WHO) suggested that the safe standard of Pb in drinking water was less than 10 µg L^−1^ [4], which was adopted by the National Drinking Water Standards for Residents in China (GB5749-2006, China). However, the U.S. Environmental Protection Agency (EPA) advised that the Pb content in drinking water must be lower than 15 µg L^−1^. Additionally, the drinking water standard of Pb in India is 50 µg L^−1^ [5]. However, in reality, the Pb content in natural bodies of water is frequently higher than the standards despite strict enforcement around the world. The researchers in [6] reported that Pb concentrations in pollution surface water and wastewater were 150 µg L^−1^ and 400 µg L^−1^, respectively, in China. Meanwhile, the highest Pb concentration of surface water was up to 2034.4 µg L^−1^ in Tamil Nadu, India [7].

Lead accumulation in water has led to serious harmful effects on aquatic and terrestrial organisms and even humans [8,9], including neurotoxicity [10], hepatotoxicity [11], and nephropathy [12]. Moreover, even trace amounts of Pb exceeding the environmental standard could harm human health. The blood–lead levels in children, due to environmental pollution, has been a global health concern. A recent survey by Bodeau-Livinec et al. [13] showed that approximately 58% of children had blood lead levels above 50 μg L^−1^ in Benin, Sub-Saharan Africa, in 2016. The U.S. Center for Disease Control and Prevention (CDC) suggested that children’s blood lead levels should be less than 3.5 μg dL^−1^; however, more than 500,000 children in the U.S. [14] and more than 120 million people worldwide have been reported with blood lead concentrations higher than 10 μg dL^−1^ [15]. Recent studies regarding the long-time exposure to Pb in humans showed that it caused arterial hypertension, cardiovascular and developmental disorders, and neurological damage [16,17]. Therefore, lead pollution in water is a significant threat to human health.

Ionic liquids (ILs) are a class of room- or low-temperature organic salts that have potential as a substitute for traditional organic solvents due to their physicochemical properties [18,19]. As environmentally friendly and “green” organic solvents, they have been widely employed in the chemical industry for organic synthesis and nanomaterials as well as in biomedicine science [20,21]. However, studies have demonstrated that ILs may not be truly green and can be toxic for organisms such as daphnid [22], earthworm [23], and fish [24]. Furthermore, recent studies have found that ILs have been detected in soil and water [25]. For example, Probert et al. [26] verified that ionic liquid 1-methyl-3-octylimidazolium (M8OI or C8mim) existed in the surrounding soils of a landfill waste site in Northumberland, England. Therefore, since ILs are solvent in water, absorbed in soil, and degraded slowly, they may be a health threat to soil and aquatic animals. 

Previous studies have demonstrated that Pb contamination in an aquatic system can cause developmental disorders and nervous system function abnormalities in juveniles. Therefore, the likelihood that lead could encounter ILs in natural bodies of water is high. If ILs promote the permeability of the blood–brain barrier, specifically for Pb, it could result in more severe neurotoxicity in aquatic animals, including fish. In the present study, a chronic co-exposure of Pb and ILs at 5% of the half-lethal concentration (LC_50_) in common carp was conducted for 28 days to determine if there was a synergistic effect from combined exposure.

## 2. Results

### 2.1. Fish Behavior

The results of the behavioral assay showed that a significant difference was observed between the control and exposed groups (Figure 1). For example, the thermal imaging and path graph of the treated fish were disproportionally active, especially when exposed to the ionic liquid M8OI, as compared to the control fish. In the Pb^2+^ group, the fish could not swim a whole lap (Figure 1A,B). In addition, the move distance, maximum speed, average speed, and absolute turn angle in the lead-treated fish were remarkably enhanced (Figure 1C–H). Fish behavioral differences were also observed between the single-exposure and co-exposure groups; for example, the fish move distance, average speed, and max distance to the border in the Pb^2+^-treatment group were higher than in the MIX group (Figure 1C,D,F) while the absolute turn angle was lower than in the MIX (Figure 1G). 

### 2.2. Neurotransmitter Dopamine

The neurotransmitter dopamine (DA) level was measured by commercial Elisa kits and is shown in Figure 2. No difference was found between the treatment and the control groups after 7 days of exposure, but the DA levels significantly increased from 14 to 28 days, with only one exception in Pb^2+^ group at 28 days.

### 2.3. Histopathological Observation

The result of the histopathological observation showed that fish exposure to Pb^2+^, M8OI, or mixed Pb^2+^ with M8OI for 28 days altered the mesencephalon structure, for example, pia mater injury, increased the gap between the stratum marginale and stratum centrale, and altered the gray matter cavity in periglomerular gray zone (Appendix A). In addition, the scale of the pia mater injury in fish in the combined-exposure group appeared slightly less than the Pb^2+^-treated fish; however, no quantitative data supported this observation (Appendix A). 

### 2.4. Transmission Electron Microscope Observation

After exposure to Pb^2+^, M8OI, or a mixture of Pb^2+^ and M8OI for 28 d, the variation of the brain ultrastructure was examined by transmission electron microscopy. As compared to the control group, the basement membrane damage and the endothelial cell vacuolation was found in all the treatment groups (Figure 3A–D). 

### 2.5. Bioaccumulation of Pb in Fish Brain

Overall, lead content in the fish brains in the Pb^2+^ or the mixture of Pb^2+^ and M8OI groups increased as the exposure time lengthened (Table 1), indicating a typical bioaccumulation of Pb in the fish brains. However, the content of Pb^2+^ in the fish brains from the single exposure group was higher than that of the MIX group at 7 and 28 d of exposure time, suggesting that the ionic liquid M8OI suppressed lead absorption rather than promoted it. 

### 2.6. BBB Permeability

The blood–brain barrier (BBB) is composed of the choroid plexus and the brain capillary walls, which inhibit hazardous substance invasions and maintain brain homeostasis [3]. In the present study, the fish brain BBB permeability was detected by Evans blue dye (EBD) fluorescence after 28 d of exposure. The EBD fluorescence intensities in all treatment groups were higher than in the control group (Figure 4A), indicating that regardless of the treatment group (i.e., Pb^2+^, the ionic liquid, or the mixture) the BBB permeability had increased. This result also suggested that the ionic liquid could not synergistically promote BBB permeability elevation by Pb^2+^ although ILs could improve membrane permeability. In addition, although no difference was observed between the Pb^2+^ and MIX groups, the EBD fluorescence in the combined group was significantly stronger than that in the M8OI group (Figure 4C). 

### 2.7. Expression of Tight Junction Coding Gene in Blood–Brain Barrier of Fish Brain

After 28 days of exposure, the expressional level of *claudin5* decreased in all treatment groups, as compared to the control group (Figure 5A); however, *claudin5* expression in the co-exposure group was higher than in the single-exposure group. Meanwhile, in the co-exposure group, *occludin* expression was higher than in the Pb^2+^ group, but lower than in the ionic liquid group (Figure 5B). A similar result was obtained when the *zo-1* expression was determined by qPCR (Figure 5C).

## 3. Discussion

Although the pollution of harmful heavy metals, such as lead, and the resultant toxicity has become a much-discussed topic, the neurotoxicity and mental retardation of children caused by lead pollution, especially in developing countries, has attracted global attention [2,7]. Studies on the effects of lead on neurobehavioral aspects, for example lead poisoning causing anxiety, hyperactivity, and other neurobehavioral abnormalities, are still limited [27,28]. In the present study, the effects of chronic exposure to lead and co-exposure of lead and the ionic liquid on the neurobehaviors of common carp were studied to uncover the mechanisms of neurotoxicity. The results of the neurobehavioral assay showed that chronic exposure to lead at lower concentrations altered fish movements and neurobehaviors, including incomplete ring motion, hyperactivity, and anxiety (Figure 1), indicating that lead exposure caused neurotoxicity in the carp. The causes of neurobehavioral changes are complex and multifaceted, including brain or neuronal damage and neurotransmitter homeostasis. Therefore, we first examined changes in the neurotransmitter dopamine levels in the brains of the carp after 28 days of lead exposure, and the result showed that dopamine levels had increased at 14 d but decreased at 28 d (Figure 2). This result indicated that lead exposure interfered with neurotransmitter homeostasis in the carp, which suggested that neurotransmitter alteration was involved in the neurotoxicity mechanism of lead. Then, we examined the structure of the carp brain by histology and transmission electron microscopy, and the results showed that lead exposure significantly altered the brain structure and caused brain cell damage in carp (Figure 3). According to toxicodynamics, lead must cross the blood–brain barrier in order to enter the carp brain. In the present study, we found that lead significantly increased the blood–brain barrier permeability (Figure 4) and entered the brain of the carp (Table 1). This result suggested that lead can cross the blood–brain barrier into the brain and cause neurotoxicity. 

Ionic liquids are a new type of organic solvents expected to replace the traditional volatile organic solvent, and, therefore, they have many potential applications in the chemical industry [18,19]. Our study indicated that chronic exposure to lead at lower concentrations caused neurotoxicity in the carp (Figure 1). Studies on the neurotoxicity of ionic liquids are still limited, although there have been many toxicological studies [29]. Our results showed that the dopamine levels in the carp brains were significantly higher than in control group after 28 days of IL-exposure, supporting the experimental results of neural behavior. Furthermore, the results of the histological and transmission electron microscopy examinations showed that IL-exposure caused brain cell injury in the carp (Figure 3C). In addition, our result showed that IL-exposure also significantly increased the blood–brain barrier permeability (Figure 4C). These results confirmed that ionic liquid is neurotoxic to carp. However, the mechanism of neurotoxicity caused by the ionic liquid on the fish should be further examined. What are the targets of ionic liquid neurotoxicity? Does the ionic liquid interact directly with the neurotransmitters, the synaptic membranes, and the ion channels, or is it only a secondary effect?

Based on the chemical properties of ionic liquids, they belong to cationic surfactants, which promote the permeability of cellular membranes. Therefore, we assumed that the ILs should have increased the blood–brain barrier permeability and synergistically facilitated the transfer of lead across the blood–brain barrier of the carp brain. On the contrary, our result showed that the IL did not promote the absorption of lead in the carp brain but, instead, inhibited it (Table 1). This unexpected result led to our speculation as to its cause. There could be a chemical reaction between Pb and IL in the exposure solution/water, so less lead was transferred into the fish brain. We determined the contents of lead and the IL in the exposure solution by HPLC from 0–24 h following exposure and also detected any new chemical reaction product by nuclear magnetic resonance (NMR). The assay results showed that the contents of both the Pb and the IL in the exposure water were stable within one day (Appendix A) and no new chemical reaction product was found (Appendix A), suggesting that they did not react chemically to form a new product. Therefore, we speculated that they could have reacted chemically, and indirectly, due to their complexation, hence reducing the absorption of lead in the carp brain. 

In summary, our results indicated that the combined exposure to lead and the IL was not synergistic; it was antagonistic to their neurotoxicity. The results of this study suggested that IL could have potential to recede the Pb-induced neurotoxicity in fish. Therefore, our study may have significance for the elimination of harmful heavy metals, such as lead.

## 4. Materials and Methods

### 4.1. Chemical

The ionic liquid M8OI (as the Cl^−^ salt, >99% purity) (CAS: 64697-40-1) was purchased from Qingdao Aolike New Material Technology Co., Ltd. (Shandong, China), and a 1.0 M of stock solution of M8OI was prepared in ultrapure water and stored at 4 °C. Lead acetate (Pb(CH_3_COO)_2_•3H_2_O, >99% purity) (CAS:301-04-2) was obtained from Aladdin Bio-Chem Technology Co., Ltd. (Shanghai, China), and the chemical reagent was dissolved in ultrapure water to prepare a 1.0 M mother solution and stored at 4 °C until use. The stock solution of chemicals was refreshed every 7 d.

### 4.2. Fish

Common carp (*Cyprinus carpio*) were purchased from the fish aquaculture farm in Yellow River Basin, Xinxiang, Henan Province, China, and temporarily cultured in the fish laboratory of the Animal Center, Henan Normal University (Xinxiang, China). The fish were fed commercial food twice daily according to weight at 26.0 ± 1.0 °C with a 14 h light/10 h dark in a recirculating aquaculture system. The experiments were approved by the Ethics Committee of Henan Normal University (HNSD-2021-07-05).

### 4.3. Exposure and Sampling 

The 5-month-old healthy common carp with body weights of 27.0 ± 3.0 g and body lengths of 13.0 ± 1.0 cm were divided into four groups (25 fish in each group): the control group (CK, culture water), Pb^2+^ group (18.3 mg L^−1^ of Pb(CH_3_COO)_2_•3 H_2_O with the effective concentration of 10.0 mg L-1 Pb^2+^), M8OI group (11 mg L^−1^ of M8OI-Cl, and the combined group MIX (18.3 mg L^−1^ of Pb(CH_3_COO)_2_•3 H_2_O and 11 mg L-1 of M8OI-Cl). The half-lethal concentration (LC_50_) values of the toxicants on the fish at 24 h were measured in the preliminary experiment, and lead acetate and M8OI-Cl LC50 were 355.15 ± 6.93 and 221.03 ± 5.10 mg L^−1^, respectively. The 5% of LC_50_ was chosen as the exposure concentration in the chronic experiment for 28 days. Each group was tested in triplicate, and the fish were fed commercial food twice daily according to weight at 26.0 ± 1.0 °C with 14 h (h) of light and 10 h of dark during exposure. Four-fifths of the exposure solution was renewed daily until the experimental termination to ensure a constant toxicant exposure concentration in the water. 

The sampling was conducted at 7, 14, and 28 days of exposure. To avoid the exposure solution entering into the fish brain when sampling, the culture water was used to clean the fish body for half an hour. Six fish were randomly selected from each group for behavioral analysis before the dissection. Then, three fish were randomly taken from each group and placed in ice water until they were unconscious, and then dissected. Three fish brains were placed in 4% paraformaldehyde (PFA) for histological examination and immunofluorescence analysis. The fish brains were stored at −80 °C until used for the subsequent experiments including RNA extraction (*n* = 3) and bioaccumulation (*n* = 6).

### 4.4. Lead Content in Common Carp Brain

In this study, the brains from the treatment and control groups were collected to measure the Pb contents. Atomic absorption spectrometry (AAS ZEEnit-700P) (Analytik Jena AG, Jena, Germany) was used to detect Pb contents by the method of sample handling, according to previous reports in our laboratory [30]. In brief, the brain tissue was dried and weighed before pre-digestion using 4 mL HNO_3_ and 2 mL H_2_O_2_ for 30 min at room temperature. Then the sample was transported to the digestion machine of Master 40 microwave oven (Shanghai Sineo Microwave Chemical Technology Co. Ltd., Shanghai, China) for further digestion and analysis.

### 4.5. Behavioral Analysis and Anxiety Identification

After 28 d of exposure, six fish from each group were randomly selected to perform the anxiety and behavioral analysis, according to a previous report [31]. Firstly, the fish were acclimatized to the environment for 30 min in a round tank, and then the fish movement trajectories and movement parameters, including movement distance, average speed, max speed, absolute turn angle, mean distance to border, and max distance to border were recorded using a HD98 camera (Hunan Huafu Technology Co., Ltd., Changde, China) for further analysis using the ANY-MAZE software (version 4.3, Stoelting, Wood Dale, IL, USA).

### 4.6. Blood–Brain Barrier (BBB) Integrity Evaluation

At the terminal of exposure, three fish from each group were chosen for BBB integrity evaluation using EBD fluorescence, according to a method previously reported by Wang et al. [32] with appropriate modifications. Briefly, each fish was sequentially placed into three tanks with different water temperatures (17 °C, 12 °C, and 8 °C) [33] until the fish lost consciousness. Then, 0.5% EBD solution (Scientific Phygene, Fuzhou, China; 40 µL g^−1^, 0.9% saline) was intraperitoneally injected into fish by microinjector. Subsequently, the fish were placed in the tank at 28 °C for recovery. Four hours later, the brains were immediately isolated from the fish and placed in 50% TCA (*w/v*) solution and homogenized and centrifuged, and the supernatants were mixed with 95% ethanol and monitored the EBD contents at 620 nm/680 nm using a fluorospectrophotometer.

### 4.7. RNA Extraction and Gene Expression Analysis

Total RNA from fish brains were extracted using an RNA extraction kit (Omega Bio-Tek, Norcross, GA, USA). The first strand cDNA was synthesized using a SuperRT cDNA Synthesis Kit (Cwbio Bio., Beijing, China). The qPCR reaction was performed by MonAmpTM SYBR Green qPCR Mix Kit (Monad Bio., Wuhan, China). The specific operation procedure and method was referenced in previous studies [34,35,36]. The GAPDH gene as an inner reference was used in the gene expression level analysis with the 2^−∆∆Ct^ method [37]. All target primer sequences were shown in Appendix A.

### 4.8. Histological Analysis

The fish brain was dissected and fixed in 4% PFA for 24 h; then, the brain was dehydrated and coated in paraffin and cut into sections at 5–10 µm thickness. The sections were dyed using hematoxylin–eosin (H&E) and put into an optical microscope for observation.

### 4.9. Transmission Electron Microscope Analysis

After 28 d of exposure, the fish brain was collected and fixed on 2% glutaraldehyde (Beijing Leagene Biotechnology Co., Ltd, Beijing, China) for transmission electron microscope (TEM) analysis. In general, the fixed brains were cut into three 1 mm thick small tissue blocks and were cleaned three times with 0.1 M phosphate buffer (pH 7.4) for 15 min each. Then, the tissue blocks were dehydrated in ascending grades of ethanol (30%, 50%, 70%, 80%, 95%, 100%, and 100% for 20 min) and 100% acetone two times for 15 min each at room temperature. The dehydrated tissue blocks were penetrated, embedded, aggregated, and cut into 60–80 nm thick flakes. Finally, the tissue flakes were stained, observed, and imaged with Hitachi HT7800 electron microscope (Hitachi, Tokyo, Japan).

### 4.10. Neurotransmitter Level Assay

Fish brain neurotransmitter level was measured by using DA Elisa kits following the commercial kit protocol (Enzyme-linked Bio., Shanghai, China) and the measure method referenced in a previous study by Shi et al. [38].

### 4.11. Data Analysis and Visualization 

The results were summarized, differences analyzed, and scientifically visualized using Microsoft Office 365 (Microsoft, Redmond, WA, USA), IBM SPSS Statistics 20.0 (IBM, Armonk, NY, USA), and GraphPad Prism 8.0.2 (GraphPad Software, San Diego, CA, USA), respectively. The results were represented as mean ± standard error of the mean (SEM). Difference analysis between the control group and treatment groups were performed with one-way analysis of variance (ANOVA) and the least-significant difference test. The “*” and “**” were expressed as *p* < 0.05 and *p* < 0.01, respectively. The difference of between the MIX group and Pb^2+^ group and M8OI group was expressed as “#” and “##” (*p* < 0.05 and *p* < 0.01, respectively).

## 5. Conclusions

Our study revealed that chronic exposure to lead even at lower concentrations caused neurotoxicity in common carp. Increases in the neurotransmitter dopamine levels and injury in the fish brains could have accounted for the neurotoxicity. Furthermore, we also found that the IL was neurotoxic to the fish although the toxicity was only mild. Of particular note, our study found that the combined exposure of lead and ILs was not synergistic but antagonistic to the neurotoxicity, suggesting that the study had significance in the elimination of harmful heavy metals such as lead, which can protect human health. 

## Figures and Tables

**Figure 1 ijms-23-06282-f001:**
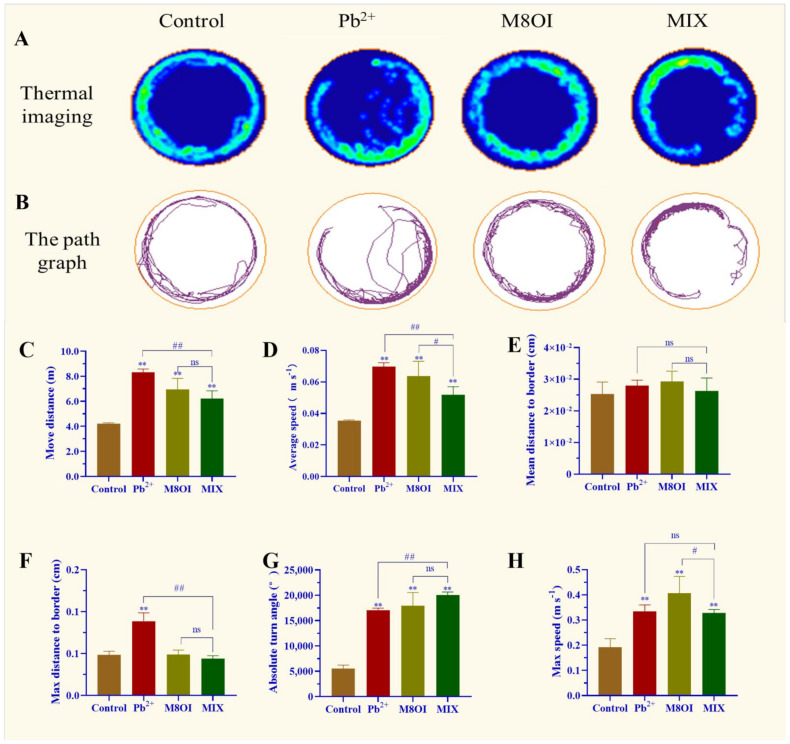
Effect of Pb^2+^, M8OI, and co-exposure on the behavior of fish. (**A**) Thermal imaging of fish activity. (**B**) The path graph of fish activity. (**C**) Move distance. (**D**) Average speed. (**E**) Mean distance to border. (**F**) Max distance to border. (**G**) Absolute turn angle. (**H**) Max speed. Data are represented as mean ± SEM. ** *p* < 0.01 is expressing the difference levels between control and exposure groups. # *p* < 0.05, and ## *p* < 0.01 are expressing the difference levels of between co-exposure and single exposure groups, and “ns” represents not significant.

**Figure 2 ijms-23-06282-f002:**
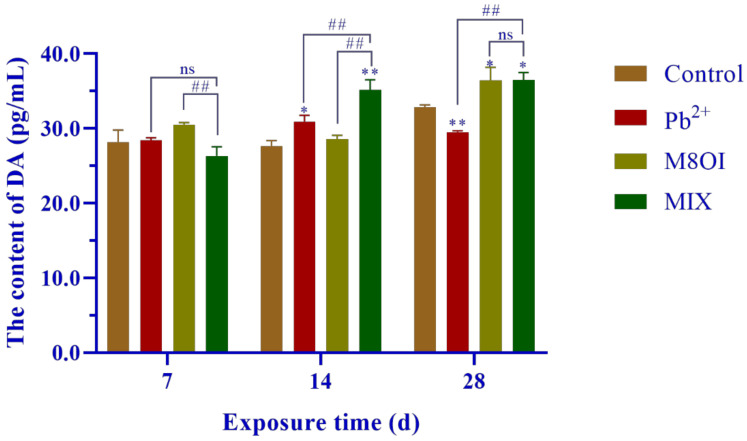
Effects of the combined exposure to Pb^2+^ and the ionic liquid M8OI on the neurotransmitter dopamine levels in fish brains. Data are represented as mean ± SEM. * *p* <0.05 and ** *p* < 0.01 are expressing the difference levels between control and exposure groups. ## *p* < 0.01 is expressing the difference levels of between co-exposure and single exposure groups, and “ns” represents not significant.

**Figure 3 ijms-23-06282-f003:**
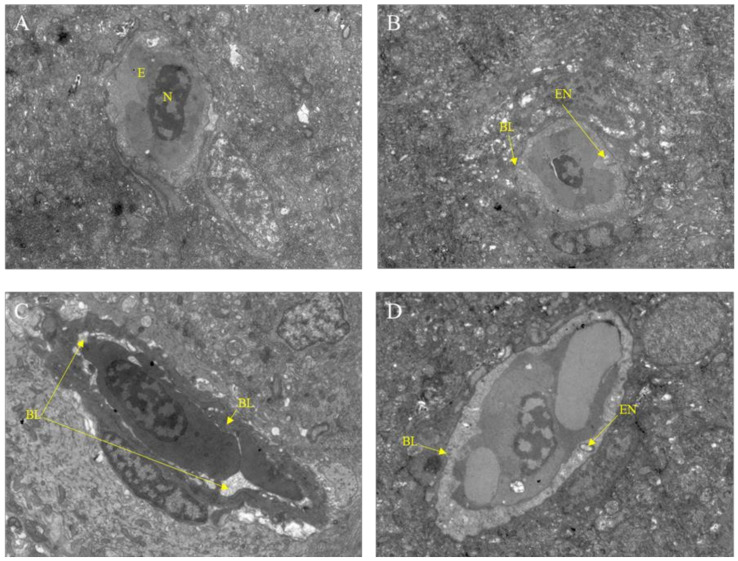
Transmission electron microscope observation of fish brain after exposure to Pb^2+^, M8OI, or mixture of Pb^2+^ and M8OI for 28 d (scale bars = 5.0 μm). (**A**) Control group; (**B**) Pb^2+^ group; (**C**) M8OI group; (**D**) MIX group. Note: E, red blood cells; N, cell nucleus; EN, endothelial cell vacuole; BL, basement membrane.

**Figure 4 ijms-23-06282-f004:**
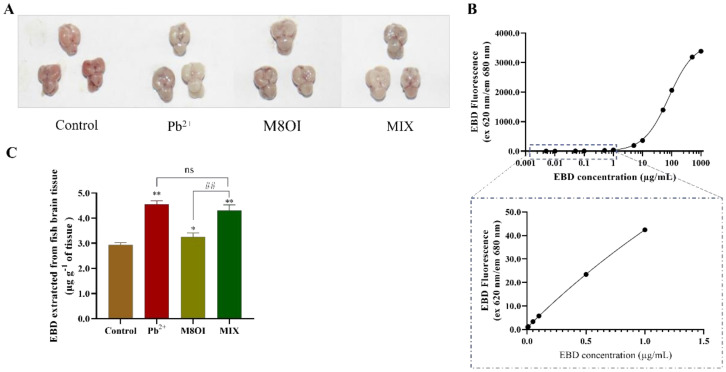
The fish BBB permeability assayed using EBD fluorescence method. (**A**) The representative pictures of EBD accumulated in brain of fish (**B**) The standard curve of EBD fluorescence was performed with 30 µL dye in 50% TCA diluted to 120 µL with 95% ethanol (*n* = 5). (**C**) The quantification and visualization of EBD content in brain on day 28 under different exposures. Data are represented as mean ± SEM. * *p* < 0.05 and ** *p* < 0.01 are expressing the difference levels between control and exposure groups. ## *p* < 0.01 is expressing the difference levels between co-exposure and single exposure groups, and “ns” represents not significant.

**Figure 5 ijms-23-06282-f005:**
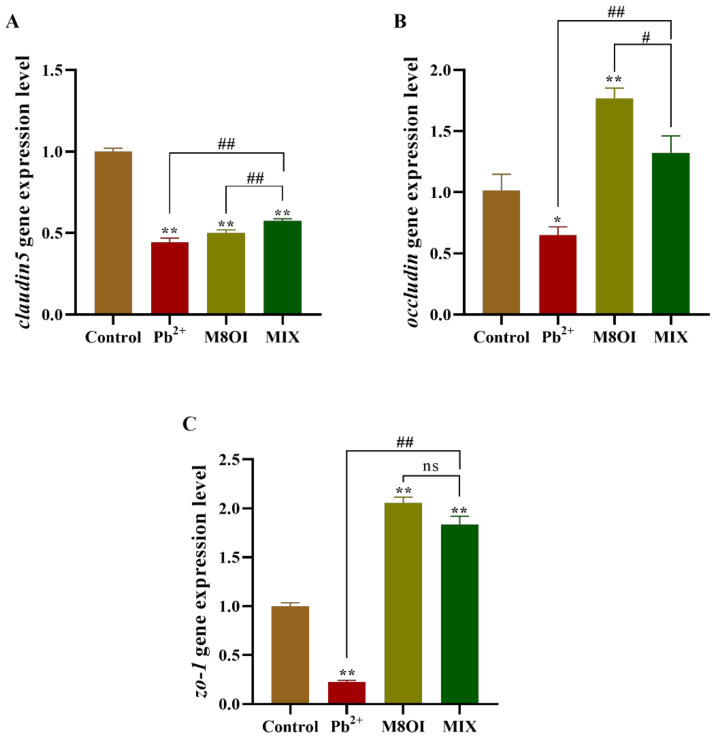
Effects of the combined exposure to Pb^2+^ and the ionic liquid M8OI on the expression of tight junction code gene in blood–brain barrier of fish brain. (**A**) *claudin5*. (**B**) *occludin*. (**C**) *zo-1*. Data are represented as mean ± SEM. * *p* < 0.05 and ** *p* < 0.01 are expressing the difference levels between control and exposure groups. # *p* < 0.05, and ## *p* < 0.01 are expressing the difference levels of between co-exposure and single exposure groups, and “ns” represents not significant.

**Table 1 ijms-23-06282-t001:** The alteration of Pb bioaccumulation in fish brains associated with exposure time.

	Exposure Time (d)	7	14	28
Pb contents (µg g^−1^ dw)	Control	-	-	-
	Pb^2+^	16.04 ± 3.16	16.47 ± 0.70	23.74 ± 2.93
	M8OI	-	-	-
	MIX	9.45 ± 1.38	12.96 ± 3.90	14.51 ± 1.91

## Data Availability

Data supporting the reported results are contained within the article.

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
