# Peer review of "Neurotoxicity of Chronic Co-Exposure of Lead and Ionic Liquid in Common Carp: Synergistic or Antagonistic?"

_ijms, 2022, doi:10.3390/ijms23116282_

Round 1
Reviewer 1 Report
This paper presents very interesting results regarding the effect of lead in conjunction with ionic liquids in the toxicity effects on carp. It is not usual to see this combination and thus the paper is suitable for publication in this journal. However the English language must be corrected before this manuscript can be considered for publication. As it is it makes reading very difficult and its hard to understand what the authors mean. Althought the experimental design seems sound and the results properly presented, I had a lot of trouble trying to understand some sections of the text and so I recommend that this manuscript be fully revised for the english language and be resubmitted for a proper and detailed evaluation.
Author Response
Please see the attachment. Thanks for your suggestion!

Reviewer 2 Report
Ding and other present a paper on exposure to lead (Pb) and an ionic liquid known as M8OI. My impression is that a large amount of experimental and analytical work went into the study, and I do not see any major issues with the methods or conclusions of the work. It is interesting that the Pb and ionic liquid are both neurotoxic, although the M8OI is less so. There is a potential antagonistic effect between them, with the result being that M8OI may reduce the impacts of Pb exposure. A very intriguing result probably worth pursuing.
The primary issue is really with the language, there needs to be some careful revisions of the english and sentence structure to increase the impact of the study. It was difficult to understand in some places.
Below are a few additional comments:
Introduction was reasonable, organization was okay but the language needs to be revised.
The US CDC has revised their blood lead reference level to 3.5 ug/dL.
Section 2.3 mislabeled header.
Figure 3. Not easy to read the red letters on the images.
I don't understand the lower and upper case "a" and "b" meaning in table 1.
Need to define "BBB" in the "BBB" permeability section.
Really liked figure 1, the movement charts were very interesting.
Not enough explanation of ionic liquids. For those of us that study Pb exposure, having more background an explanation on ionic liquids and their uses would help the reader understand the significance. How widespread are these ionic liquids? How does their presence in the environment compare to lead, i.e., are they expected to occur in the same places?
Says the IL could be used to treat lead exposure, but also that IL are neurotoxic? I'm not sure if that is the intention, but it seems odd to suggest this if the results indicate that IL are neurotoxic.
Unclear how the Pb contents were measured in table 1 in the paper. The methods are in the supplementary file, but I don't think this was referred to in the paper.
Don't need to say "In conclusion" in the conclusion section.
Methods should come after intro, before results and discussion.
Unclear why some methods were in the main text and some in the supplementary file.
Figure S4 in the supplementary file was pretty dark, not sure what I was meant to see.
Author Response

(The authors gave the same response as above.)

Reviewer 3 Report
The manuscript does not include the material and methods, so the quality of the results is in doubt. This is the most important section, because the lack of material&methods and quality parameters is a big problem.
Author Response

(The authors gave the same response as above.)

Round 2
Reviewer 3 Report
The manuscript now is complete.